# Decoding gripping force based on local field potentials recorded from subthalamic nucleus in humans

Huiling Tan[1,2]*, Alek Pogosyan[1,2], Keyoumars Ashkan[3], Alexander L Green[2], Tipu Aziz[2], Thomas Foltynie[4], Patricia Limousin[4], Ludvic Zrinzo[4], Marwan Hariz[4], Peter Brown[1,2]

[1]Medical Research Council Brain Network Dynamics Unit, University of Oxford, Oxford, United Kingdom; [2]Nuffield Department of Clinical Neurosciences, John Radcliffe Hospital, University of Oxford, Oxford, United Kingdom; [3]Department of Neurosurgery, Kings College Hospital, Kings College London, London, England; [4]Sobell Department of Motor Neuroscience and Movement Disorders, UCL Institute of Neurology, London, United Kingdom

**Abstract** The basal ganglia are known to be involved in the planning, execution and control of gripping force and movement vigour. Here we aim to define the nature of the basal ganglia control signal for force and to decode gripping force based on local field potential (LFP) activities recorded from the subthalamic nucleus (STN) in patients with deep brain stimulation (DBS) electrodes. We found that STN LFP activities in the gamma (55–90 Hz) and beta (13–30m Hz) bands were most informative about gripping force, and that a first order dynamic linear model with these STN LFP features as inputs can be used to decode the temporal profile of gripping force. Our results enhance the understanding of how the basal ganglia control gripping force, and also suggest that deep brain LFPs could potentially be used to decode movement parameters related to force and movement vigour for the development of advanced human-machine interfaces.

*For correspondence: huiling. tan@ndcn.ox.ac.uk

Competing interests: The authors declare that no competing interests exist.

## Introduction

Accurate control of grip force is essential in the manipulation of objects in everyday life. Knowledge of how gripping force is encoded in the brain would facilitate the design and control of brain machine interfaces (BMI) driving neuroprosthetics to help physically impaired patients. However, the results of studies aimed to decode force based on cortical neural activity are still far from consistent and satisfactory, and no BMI user has yet achieved manipulation of the force generated by a robotic hand (*Velliste et al., 2008*; *Collinger et al., 2013*), or the control of the simulated grasp force used for a virtual object (*Bensmaia and Miller, 2014*).

Motor areas of the basal ganglia have long been associated with the scaling of motor vigour, measured in terms of the amplitude and speed of a movement or gripping force, although this is certainly not likely to be their exclusive function (*DeLong and Wichmann, 2010*). Neuronal recordings in monkeys and imaging studies in healthy humans have suggested that the basal ganglia play an important role in the control of the scaling of motor responses (*DeLong et al., 1984*; *Turner and Anderson, 1997*; *Spraker et al., 2007*; *Vaillancourt et al., 2007*). Direct recordings from basal ganglia targets in patients suggest that changes in frequency specific activities in the local field potential (LFP) contribute to the selection of effort or force levels for voluntary movements. For example, the power over the gamma band (60–80 Hz) in the LFP in the globus pallidus correlates with the movement amplitude and velocity of the contralateral hand of patients with cranial dystonia

**eLife digest** The basal ganglia are a group of structures deep within the brain. Alongside its many other roles, it is thought to be able to control the vigour of movements, including how quickly we move and how much force we use to grip objects. Some of the best evidence for this comes from patients with Parkinson's disease, who show abnormal activity of the basal ganglia. These patients move more slowly than healthy individuals and often struggle to grip objects with desired force, making it difficult to perform everyday tasks.

Inserting electrodes into the brain and using them to electrically stimulate the basal ganglia is one of the most effective treatments for severe Parkinson's disease. To examine the relationship between activity in the basal ganglia and grip strength, Tan et al. studied activity in the basal ganglia of patients as the individuals attempted to grip an object. On each trial specific types of coordinated activity within the basal ganglia predicted when movements would start and how much force the person would use.

By constructing a mathematical model of the data, Tan et al. showed that the coordinated activity of cells within the basal ganglia indirectly controls motor vigour. The model also suggested how this process might go wrong in Parkinson's disease. Activity in the basal ganglia predicts grip strength with such accuracy that it might even be possible to exploit this relationship to help individuals with paralysis. If imagining a movement triggers the same basal ganglia activity as performing it, patients could in principle use this activity to control robotic devices rather than limbs.

To test this idea, future work should examine whether the features of basal ganglia activity can drive a Brain Machine Interface for robotic control in real-time. If so, the next question is whether signals from the basal ganglia are sufficient to control robotic devices, or if signals from other parts of the brain are needed too.

(*Brücke et al., 2012*). Similar correlations have been noted in patients with Parkinson's disease between movement speed and the power in the gamma band in the LFP picked up from the STN (*Joundi et al., 2012*). Our previous studies also showed that suppression in the beta band (13–30 Hz) and power increase in the gamma band of the STN LFP may correlate with forces or efforts made over the lower and higher effort ranges, respectively, in a manner independent from the effector that was activated (*Tan et al., 2013*, *2015*). These results suggest that the signals from basal ganglia may serve as a central signal indexing motor effort, which in turn modulates force in manual grips. However, most previous studies are based on static linear correlations and averaged data; the dynamic relationship between activities of different frequencies in the basal ganglia LFP and generated force, and whether this relationship can be used to decode gripping force based on basal ganglia LFP signals has not been investigated on a trial by trial basis.

The aim of the current study was to decode gripping force profiles from LFPs recorded in the STN. We hypothesized that beta and gamma band activities will be the most informative features in predicting the force profile generated by the contralateral hand, and that a simple first order linear dynamic model is sufficient to capture the relationship between STN LFP features and generated force. Our results suggest that reciprocal changes in synchronised oscillatory population activity in different frequency ranges provide potential control signals for the motor plant, the action of which can be modelled as a first order linear dynamical system. At the same time our results raise the possibility of using the LFP signal recorded from deep brain structures to provide stable and high-performance control signals for BMI driven neuroprosthetic grasping in paralysed patients.

## Results

In the core study, patients with idiopathic Parkinson's disease who underwent implantation of DBS electrodes into the STN were asked to grip a dynamometer with different effort levels. Subjects were instructed to respond 'in their own time'. For each patient, both hands were tested in separate sessions, with 31 ± 2 grips per hand. Local field potentials (LFP) from the DBS electrodes and the gripping force measured by the dynamometer were simultaneously recorded (see details in Materials and methods). *Figure 1A* shows the measured force trajectory from one typical subject.

*Figure 1B* shows the average power spectrum for the group of recordings (n = 9) with significant movement-related modulation in both beta and gamma bands during gripping. There is a movement related gamma increase and beta power suppression in line with previous reports (*Cassidy et al., 2002*). This pattern is compared to the other group (n = 9) in which no significant movement-related modulation was observed in either the beta or gamma band in *Figure 1C*. In the latter group the mean spectrum demonstrated increased activity (synchronisation) at low frequencies, extending to 25 Hz, especially during the force onset phase. A possible cause for this low frequency power increase at the time of movement onset, which then contaminated the beta band, is movement related artefact. The latter is overshadowed by beta band desynchronization in the recordings comprising *Figure 1B*.

*Figure 1D and E* show the normalized force trajectories with different self-rated effort (SRE) levels averaged across hands in the two groups. *Figure 1G and H* show the trajectories of force yank (the differentiation of force against time) with different SRE for the two groups. In the first group which showed modulation in the beta and gamma band, the average force during the holding phase (1.0–2.0 s after cue) and the peak force yank in the force initialisation phase scaled with SRE in all individual hands. The Spearman correlation coefficient between stable force and SRE ranged between 0.6454 and 0.9728 (median value = 0.9409, *Figure 1F*); and the correlation coefficient between peak force yank and SRE ranged between 0.4514 and 0.9341 (median value = 0.8688, *Figure 1I*). In the second group, the stable force did not scale with SRE, with abnormally increased force and force yank at lower effort levels, compared with the first group, as shown in *Figure 1E and H*. The correlation coefficient between the stable force and self-rated effort in the second group (ranged between 0.0258 and 0.8334, median value = 0.6023) was significantly lower than in the first group (t(16)=5.386, p<0.0001 with un-paired t-test applied to Fisher transformed r values, *Figure 1F*). The correlation coefficient between the peak force yank and SRE in the second group (ranged between −0.4992 and 0.6944, median value = 0.3791) was also significantly lower (t(16) =4.977, p=0.0001 with un-paired t-test applied to Fisher transformed r values). This difference in the scaling ability may account for some variations we observed in LFP reactivity. Thus, patients in the second group exhibited impairment in the scaling of force with effort. The lack of beta and gamma reactivity during gripping in this group would be consistent with a role for these activities in the coding of effort in to force. The deficit in the scaling of force with effort may be due to disease related differences between the groups, or due to the temporary post-surgical stun effect which may compromise STN function in some patients.

The first group of 9 STN sides showing significant movement-related modulations in both bands were selected for further analysis as (1) this group showed normal scaling of force and force yank with effort; (2) movement-related modulations were presumed to indicate recordings made in or near the dorsal STN; (3) significant movement-related modulations were taken as evidence of relatively physiological functioning given that the motor reactivity of the two rhythms is increased by dopaminergic medication (*Anzak et al., 2012*), and (4) lack of movement related modulations might indicate a temporary surgical stun effect and temporary damage to the STN (*Chen et al., 2006*) or targeting variance. No other clinical details of the patients (shown in *Table 1*) explain the difference between the two groups.

## Force decoding based on features from STNr LFP and a linear dynamic model

Our main interest was the initialisation, development and the average force during the 'holding phase' of each grip; therefore we focused on the period of time from one second before the cue to 2.8 s after the cue (before force releasing) for decoding. The force trajectory of each individual trial of each subject was normalized against the average maximal force that subject achieved in their maximal effort trials. We hypothesised that the relationship between LFP features and generated grip force (the transfer function) could be captured by a first order linear dynamic model (*Equation 1*):

$$Force = LFP * \frac{K_p}{T_p \cdot s + 1} e^{-T_d \cdot s} \tag{1}$$

Where $K_p$ is the steady state proportional gain, $T_p$ is the time constant of the first order system

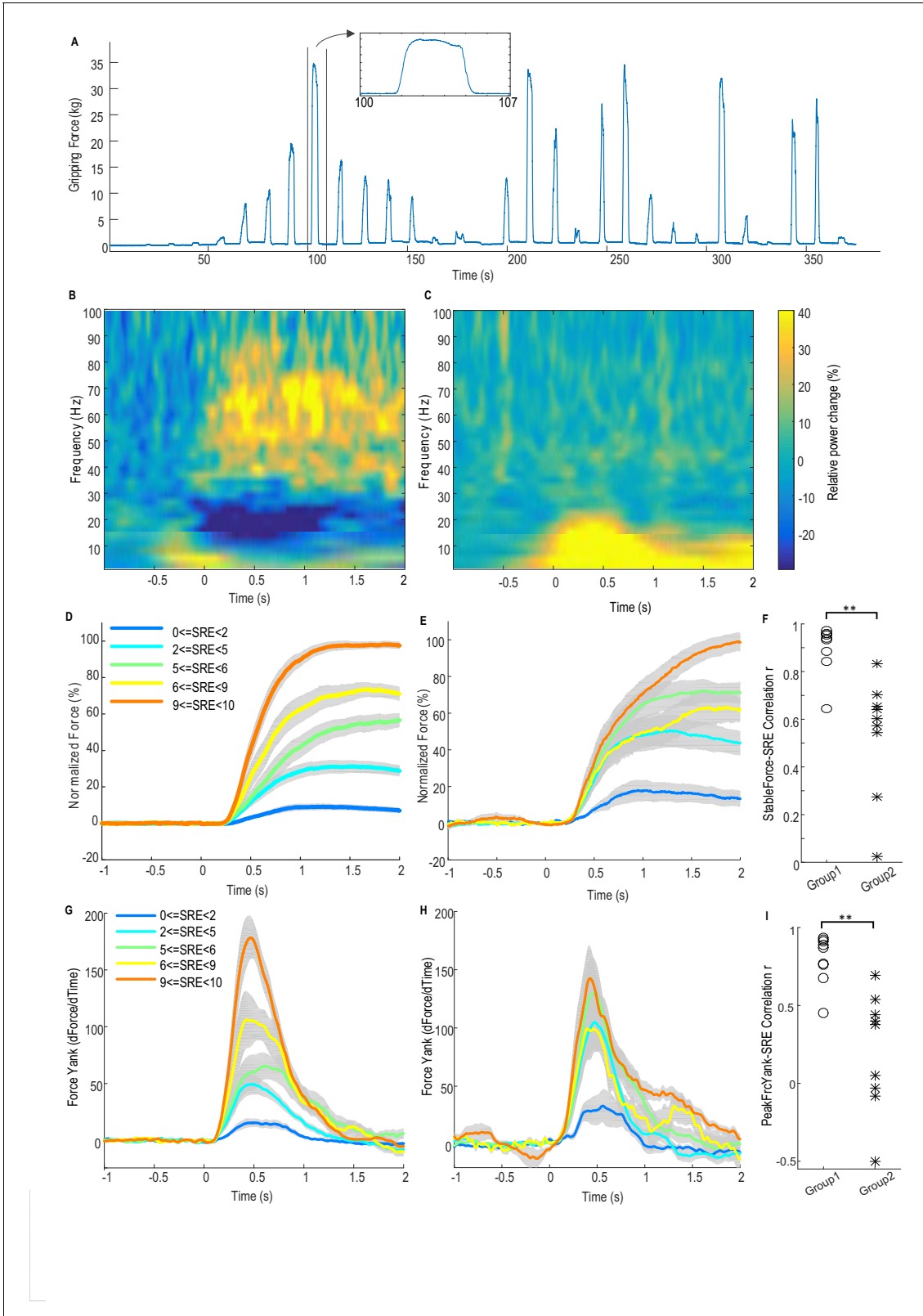

**Figure 1.** Force-effort scaling and spectra of average power changes relative to pre-movement baseline for two groups of electrodes. (A) Trajectory of measured force from one exemplar subject. (B) From one group of electrodes (n = 9), a significant reduction of power in the beta band (13–30 Hz) and increase in power in the broad gamma band (55–90 Hz) was observed during gripping. (C) In another group of electrodes (n = 9), significant simultaneous modulation was absent in the beta and gamma band with movement, and there was instead an increased power across the low frequency

*Figure 1 continued on next page*

*Figure 1 continued*

band during gripping. Trajectories of force (**D**) and force yank (**G**) for Group one show that the stable force during the holding phase (1–2 s after cue) as well as the peak force yank in the force initialisation phase scaled well with self-rated effort (SRE). In group 2, the stable force (**E**) and force yank (**H**) did not scale with effort as well as Group 1. Group two had significantly lower correlation coefficients between stable force and SRE (**F**) and between the peak force yank and SRE (**I**) compared with group 1 (p<0.0001), indicating some impairment in the scaling of force with effort. Time 0 indicates the onset of the cue to start a grip in **B–E**, **G** and **H**. Note that data from three electrodes are excluded. Of these, two had significant modulation in the beta band but not in the gamma band, and one had significant modulation in the gamma band, but not in the beta band.

The following source data is available for figure 1:

**Source data 1.** The Matlab data file containing source data related to *Figure 1*.

which is a measure of how fast the force output responds to brain signal changes, and $T_d$ is the time delay between the brain control signal (LFP) and the measured force which describes the latency between the timing of brain signal changes and force output onset. Different models with different assumptions about what is the effective force control feature from the STNr (STN region) LFP and how different features are combined to encode force were tested (see *Table 2* and details in Materials and methods).

Cross-validation, within each STN/contralateral hand as well as across different patient groups, was used to evaluate the generalisability and decoding performance of the proposed models for predicting gripping force (see details in Materials and methods). Within each STN/contralateral hand, after the model parameters had been fitted based on training data, several key variables were quantified to evaluate the performance of the model in predicting force in another set of test data recorded from the same hand: (1) the correlation coefficient between the predicted force and measured force (WithinTrialR), which can be used to evaluate the prediction accuracy of the force development and force trajectory within each individual trial. (2) The root mean square error, which quantifies the average distance between the prediction and actual measurements normalized by the maximal measured value (nRMSE). (3) The correlation coefficient between the average predicted force at the holding phase (over 1–2 s after cue onset, when the gripping force was relatively stable) and the measured force at the holding phase across different trials (StableFrcR). This was used to evaluate the prediction accuracy of the stable target force across different effort levels. (4) The difference between the timing of the predicted force onset and the timing of the measured force onset (DifRT).

## Decoding the average force over multiple trials

The performances of different models in predicting force averaged across multiple trials using different STN features (see Materials and methods for details) were first evaluated. To achieve this, gripping force and LFPs measured from the contralateral STNr were grouped into low effort trials (with self-reported effort <= 5, trial number = 15±1) and high effort trials (with self-reported effort > 5, trial number = 14±1). Average STNr LFP features and force trajectories were calculated for each effort condition. The STNr LFP features and force for one effort condition were used to estimate the model parameters (model fitting), and the models were then used to predict force for the other effort condition (model testing). The within-trial correlation coefficients between the predicted force and measured force (WithinR) and the nRMSE of the predicted force from different models during model testing are shown in *Figure 2A and B*. There was no significant difference consistent across the STNs between the three models using beta and gamma ERS as model inputs (*Models 1–3*) in the predictive performance in terms of either WithinR or nRMSE. These three models with both beta and gamma activity as inputs performed better than models in which extra information about alpha activity was also included (*Models 4,5*), or models with activities from a single frequency band as model input only (*Models 6–8*). *Figure 2C* shows the BIC values combining the force prediction for low effort and high effort for all models. One-way repeated ANOVA identified a significant effect of models in the BIC (F(7,56)=11.777, p<0.001). Paired t-tests showed that *Model 2* was significantly better, in terms of BIC, than *Models 4–8* after FDR multiple comparison correction: (t(8)=−3.63, p=0.007 compared with *Model 4*; t(8)=−4.950, p=0.001 compared with *Model 5*; t(8)=−9.347,

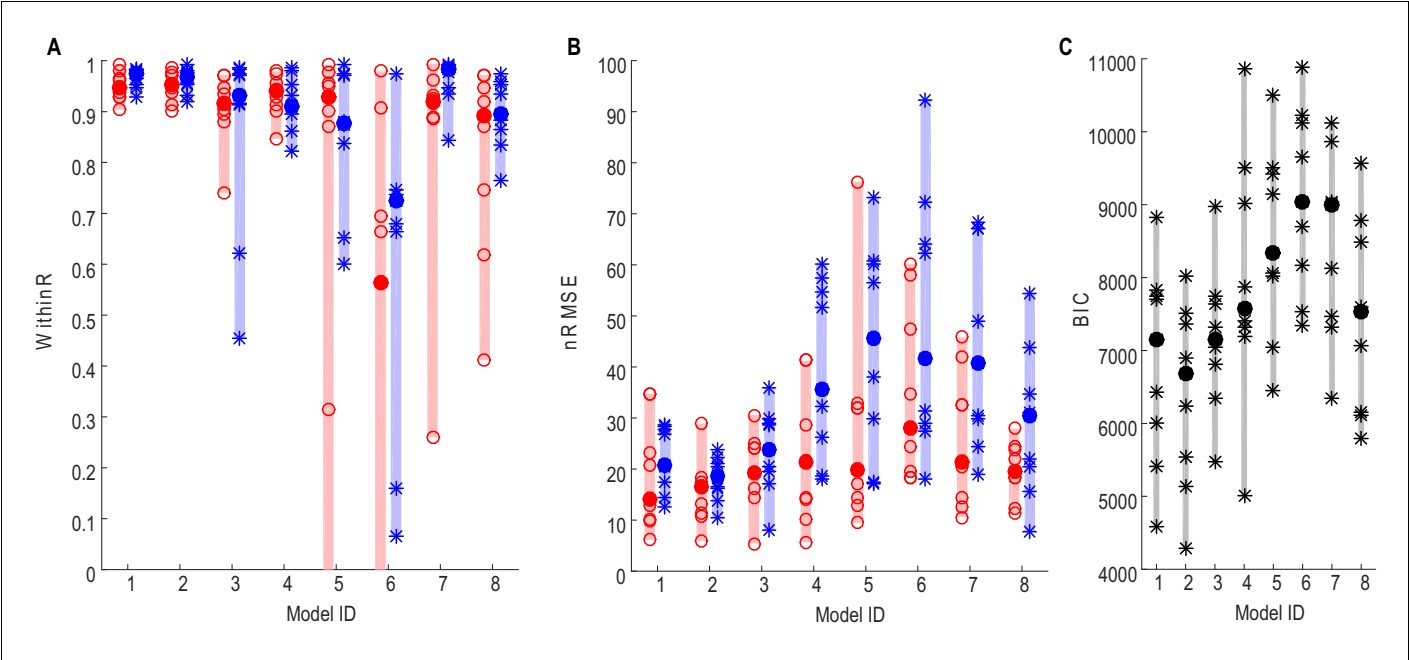

**Figure 2.** Force prediction performances of different models evaluated in terms of within-trial correlation (**A**), RMSE (**B**) and BIC (**C**). The filled dots and shaded bars show the median and range across all STNs; the open circles and stars show the data for each individual STN. The red dots and bars show performance in predicting high effort forces, while using data from low effort trials for model fitting; the blue dots and bars show performance in predicting low effort forces, while using data from high effort trials for model fitting. (**C**) The total BIC values combing the force predictions for low effort and high effort for all tested models. The filled dots and shaded bars show the median and range across all STNs; the stars show the data for each individual STN (some overlap). *Models 1–3* use beta and gamma ERS as model inputs; *Models 4–5* use activities from all three frequency bands (alpha, beta and gamma) as model inputs; *Models 6–8* use activities from a single frequency band (alpha, beta and gamma, respectively) as model inputs.

The following source data is available for figure 2:

**Source data 1.** The excel data file related to *Figures 2* and *6*.

p<0.001 compared with *Model 6*; t(8)=−5.757, p<0.001 compared with *Model 7*; t(8)=−3.675, p=0.007 compared with *Model 8*).

*Figure 3* shows how *Model 2* with beta and gamma having different linear gains can be used to predict the force generated by the contralateral hand for individual STNs. In *Figure 3A*, data averaged across low effort trials (with self-reported effort <= 5) were used for the estimation of model parameters (model fitting). The predictive performance of the model was evaluated on the data averaged across high effort trials for the same STN (with self-reported effort > 5, *Figure 3B*). *Figure 3C* shows the fitting and predictive performance of the model on data from the other 8 STNr electrodes in which consistent movement-related power modulations in both the beta and the gamma band were observed. The correlation coefficient between the predicted force and the measured force across all STNs was between 0.902 and 0.987 with a median value of 0.952 for high effort force (data during low effort used for model parameter estimation) and between 0.921 and 0.992 with a median value of 0.969 for low effort force (data during high effort used for model parameter estimation). The nRMSE of the prediction force was between 5.60% and 28.07% with median value of 16.5% for high effort levels and between 10.5% and 23.8% with median value of 16.4% for low effort levels. The differences in reaction times (DifRT) between the predicted force and the measured force were 11 ± 27 ms for high effort levels and −26 ± 31 ms for low effort levels. These were not significantly different from zero (t = 1.25, p=0.247 for high effort levels and t = 1.03, p=0.333 for low effort levels, one-sampled t-test compared to zero). The estimated Td ranged between 0 and 264 ms for low effort trials and ranged from 0 and 277 ms for high effort trials across STNs. Together, the results demonstrated that a simple first order linear dynamic model with beta

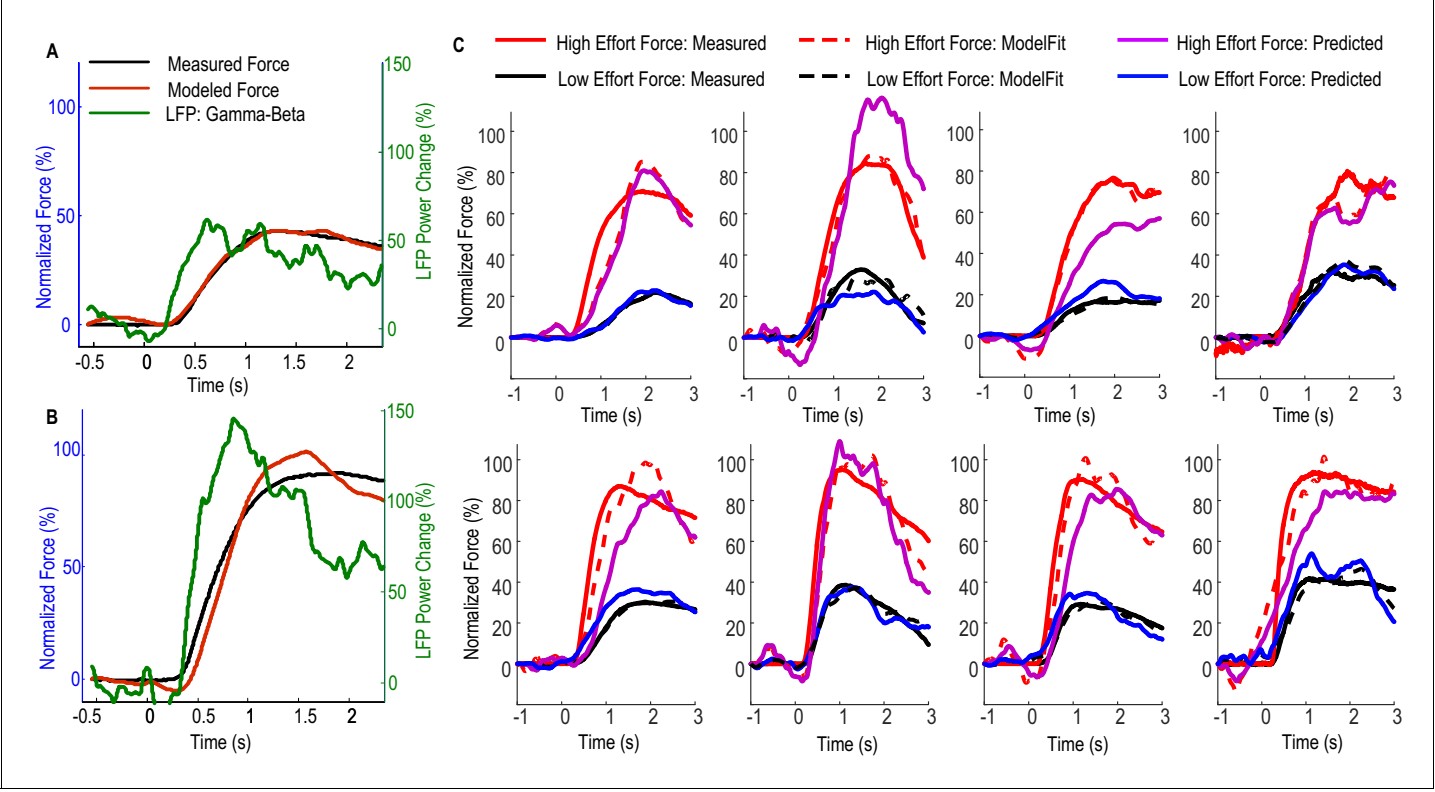

**Figure 3.** Fitting and predicting performance of the model for predicting force averaged across multiple trials. (**A**) The fitted model based on data from low effort trials for one exemplar STN and the contralateral hand. (**B**) The fitted model was used to predict the average force for high effort trials for the same STN and contralateral hand. (**C**) The fitted (dashed lines) and predicted force were compared against the measured force for the other 8 STNs in which consistent movement-related modulations in both beta and gamma bands were observed. Predicted force traces for high effort trials were derived from the model fitted to data from low effort trials and vice versa. Time 0 indicates the onset of the cue to start a grip in all plots.

The following source data is available for figure 3:

**Source data 1.** The Matlab data file containing source data related to *Figure 3*.

and gamma as inputs with different linear gains can be used to describe the relationship between STNr activity change and measured force; and that, in addition, the model with parameters derived from one set of data can be used to predict force exerted at other effort levels in the same subject.

## Decoding gripping force of individual trials and the laterality in force decoding

*Figure 4* shows the performance of the model with beta and gamma power change in the STNr LFP as inputs (*Model 2*) in predicting contralateral gripping force in individual trials for one exemplar STN. For half of all the individual trials (n = 21) recorded from this STN and contralateral hand, the within-trial correlation coefficient between the predicted force and measured force was equal or larger than 0.78. The actual measured stable force during the holding phase and the predicted stable force were then quantified for each individual trial (*Figure 4D*). *Figure 4D* shows that the stable force during the holding phase varied from zero to 100% of the maximal voluntary force across all the trials. Linear correlation was applied to the actual measured stable force during the holding phase and the predicted stable force of all individual trials. This predicted stable force correlated with the measured stable force (n = 21, r = 0.815, p<0.001), suggesting that the STNr LFP in conjunction with the dynamic model can predict both the force trajectory in individual trials and the stable force achieved across trials with different effort. The line of best fit between the predicted stable force and measured stable force was y = 0.96x + 2.4. The regression gradient close to unity suggests that the prediction matches well with the measurement with no systematic overestimation or

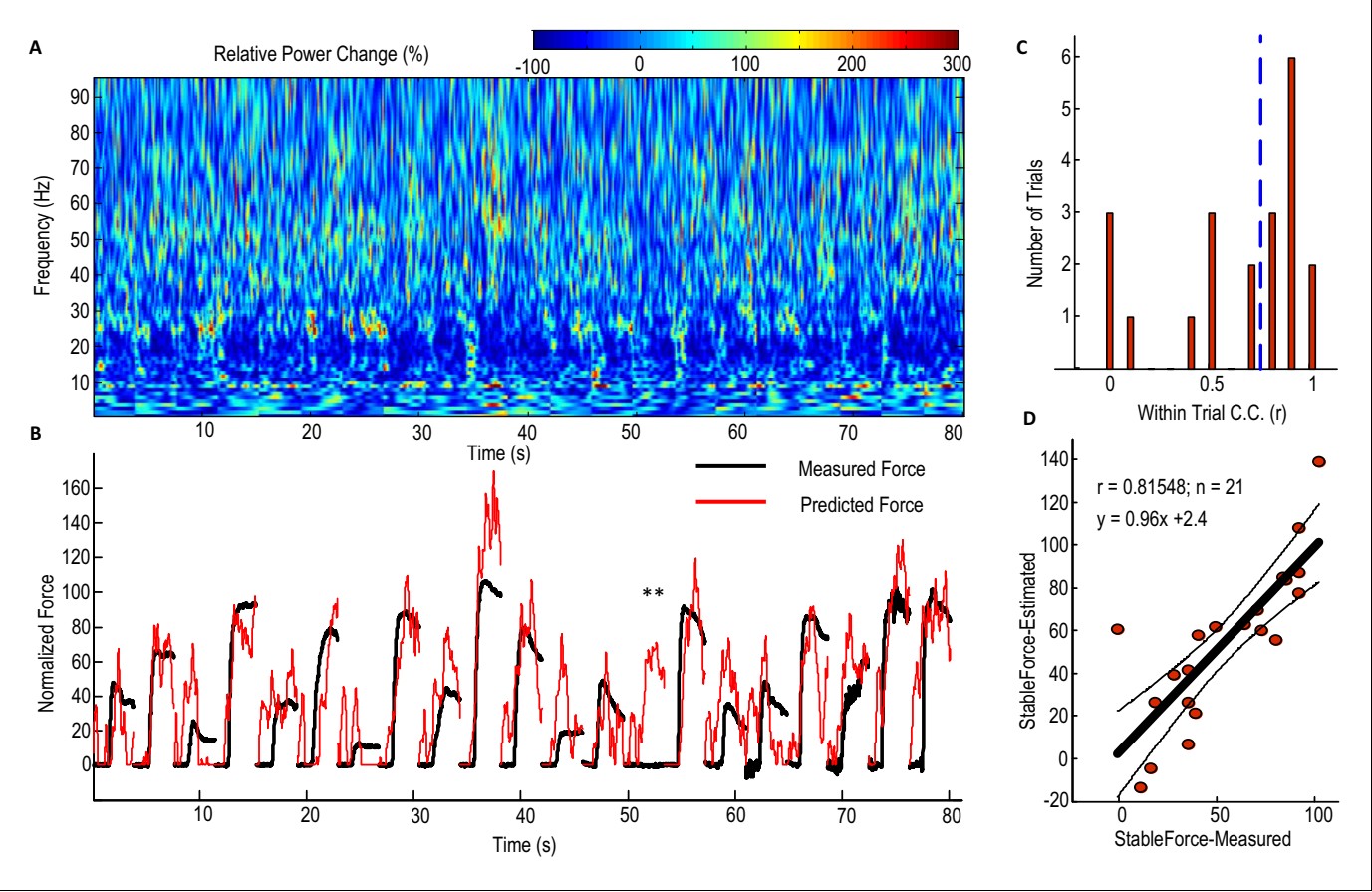

**Figure 4.** Predicting force profile of individual grips based on beta and gamma activities from STN LFP (one exemplar subject). (A) Time-evolving power spectrum of the bipolar STN LFP channel used for decoding force. (B) The predicted force (in red) compared with the measured force (in black). ** indicates the trial where STN LFP predicted increased force but with no measured force from the dynamometer. Grips are concatenated in A and B. (C) Distribution of the within-trial correlation coefficient (WithinR) between predicted force and measured force, with the dashed blue line the median value of the WithinR for all trials. (D) Scatter plot between the predicted stable force (average force during the second of holding phase) and measured stable force for all tested trials. The correlation coefficient between the predicted and measured stable force across trials was 0.815 for this subject. The regression slope of 0.96, which is close to 1, shows that there is no systematic under-estimation. The black lines show the regression line and 95% confidence interval.

The following source data is available for figure 4:

**Source data 1.** The Matlab data file containing source data related to *Figure 4*.

underestimation. During one trial for this patient, there is a force increase predicted from the STNr LFP, but there is no actual measured force change in the dynamometer (indicated by ** in *Figure 4B*). This may be due to changes in the STNr LFP signal following the cue without the actual force change being registered in the dynamometer, as might arise when movements of the limb did not involve the dynamometer.

Considering all the STNs in which significant movement-related modulations were observed in both beta and gamma bands (N = 9), the correlation between the predicted stable force and measured stable force across trials ranged from 0.448 to 0.913 for the contralateral hand when *Model 2* was used, with the regression gradient between the predicted stable force and measured stable force ranging from 0.91 to 1.12 (not significantly different from unity across STNs based on one-sampled T-test, p=0.431). There was no significant difference between the three models which took into account both beta and gamma activities (F(2,16)=0.207, p=0.815, *Figure 5A*). In addition, the distribution of the within-trial correlation coefficients across all trials was not significantly different between *Model 2* and *Model 1* (ks-test p=0.2850), *Model 1* and *Model 3* (ks-test p=0.9584) or

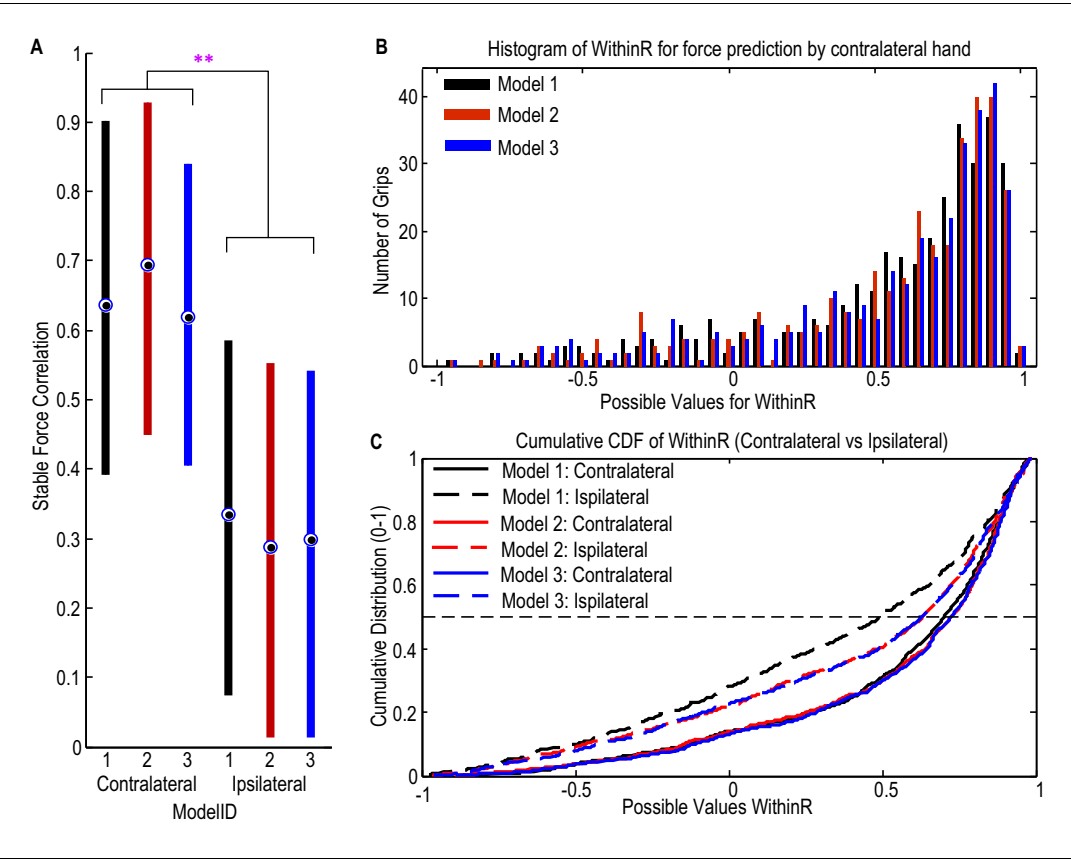

**Figure 5.** STN LFP features predict gripping force profile generated by the contralateral hand. (**A**) The correlation coefficients between the measured stable force and the predicted stable force were higher for the force generated by the contralateral hand than that by the ipsilateral hand. There was no significant difference when different models based on both beta and gamma activities from STN LFP were used. The dots and bars show the median value and the range of values for different STNs. ** indicate a significant difference in the prediction performance when the LFPs from the ipsilateral STN was used for decoding. (**B**) The histogram of the within-trial correlation coefficients between predicted force and measured force (WithinR) for the contralateral hand considering all the trials and all the STNs. (**C**) Cumulative distribution function (CDF) of the WithinR for the force generated by the contralateral hand (solid lines) and the ipsilateral hand (dashed lines). The CDF indicates the probability that WinthinR has a value less than or equal to a certain value on the x-axis. Data presented in this figure are for all the STNs in which significant modulations were observed in both the beta band and gamma band.

The following source data is available for figure 5:

**Source data 1.** The Matlab data file containing source data related to *Figure 5*.

*Model 2* and *Model 3* (ks-test p=0.9228). For convenience therefore, *Model 2* was taken as representative of *Models 1–3*. Considering all the individual trials from the 9 STNr, the median value for the within-trial correlation coefficient was 0.732 for *Model 2* (*Figure 5B and C*). There were around 13% of trials during which the correlation between the measured force and the predicted force were negative, suggesting a failure in predicting force trajectory.

The predictive performance of the STNr LFP for the force generated by the ipsilateral hand was also evaluated. The cross-trial correlations between the predicted stable force and measured stable were significantly lower for the ipsilateral hand than for the contralateral hand (F(2,16)=10.629, p=0.012). The distributions of WithinR for ipsilateral force prediction were significantly different from that for the contralateral force prediction (ks-test p<0.001 no matter whether *Model 1*, *Model 2* or *Model 3* was used), indicating that the STNr LFP was more informative in predicting force generated

by the contralateral than the ipsilateral hand. However, the degree of lateralisation could have been underestimated due to the potential presence of mirror movements, even though we asked patients to avoid these and in half those recordings used for decoding we also had bilateral upper limb EMG recordings.

## Factors affecting the decoding performance of the STNr LFP

*Figure 6* shows how the performance of force prediction based on the STNr LFP changes with movement related reactivity in the gamma band and beta band. Here we considered all the STNr with significant movement-related modulations, whether either in one or other, or both frequency bands of interest (N = 12). The median values of the within-trial correlation coefficients and stable force correlation coefficients increased with the average movement related synchronisation in the gamma band, with an exponential model $y = a * e^{-bx} + k$ explaining 75.7% (p<0.001) of the variance in WithinR and 88.1% (p<0.001) of the variance in StableR across STNs. There was also a trend for better prediction of force with increasing movement related beta desynchronization, but linear regression fitting was not significant for either WithinR or StableR. If we assume that movement-related synchronisation in gamma activity upon gripping is a good proxy for proximity to the motor region of the STN (and perhaps the upper limb representation within it) then these findings suggest that the electrode has to be very near to this region if recorded activity is to have a decent prospect of force prediction. This assumption was borne out by the fact that the movement-related synchroni-sation in gamma activity dropped by 71 ± 7.7% from the bipolar channel showing the most move-ment-related modulation to the average modulation in the remaining two bipolar channels. This drop was not so acute, 59 ± 7.9%, for movement related modulation in the beta band, which also only showed a trend for better prediction of force as movement related modulation in this band increased. Finally, there was a lack of significant correlation between the accuracy of the prediction of force and baseline beta (Spearman r = 0.468, p=0.13 for WithinR; Spearman r = 0.363, p=0.25 for StableR) or gamma activity (Spearman r = 0.281, p=0.37 for WithinR; Spearman r = 0.273, p=0.39 for StableR) measured during rest across subjects, suggesting that it is the reactivity of the power changes during gripping that is important for prediction, perhaps because it is more specific for the corresponding motor representation.

## Cross validation of the force prediction models in another independent patient group

Finally, the first order dynamic model with beta and gamma power changes as inputs (*Model 2*) was used to predict the gripping force from individual trials in an independent patient group that per-formed a different, but related, paradigm. Ten patients were recorded in this study and were asked to grip as fast and as strongly as they could in each trial in response to an external cue. Twenty trials were collected for each hand when the patients were on their normal dopaminergic medication. Details of patient information and experimental paradigm were previously reported (*Anzak et al., 2012*). The average movement related power change in the STN LFP activities and results of force prediction in this patient group are shown in *Figure 7*. The median correlation coefficient between predicted force and measured force for individual trials (median WithinR) ranged between 0.3848 and 0.9421 for the 20 STNs and contralateral hand (*Figure 7B*). Considering all the 397 individual tri-als across all STNs, more than 50% of the trials had WithinR more than 0.7859 (*Figure 7C and E*). We also found that incorporating alpha activity (*Model 4*) improved the force prediction accuracy for maximal effort gripping in this patient group. When *Model 4* was used, the median WithinR ranged between 0.6731 and 0.9625 across all STNs, and more than 50% of all individual trials had WithinR more than 0.8699. BIC analysis showed that *Model 4* (BIC = 2116.2±135.7) was significantly better than *Model 2* (BIC = 1873.0±110.3) for this patient group performing maximal effort gripping (ΔBIC = 243.3±95.9, t(19)=2.5369, p=0.020 comparing the BIC values, *Figure 7D*).

## Discussion

Here we demonstrate that the LFP signals recorded from the STNr, which is a target of deep brain stimulation (DBS) for movement disorders such as the Parkinson's disease, can be used to decode the gripping force made by the contralateral hand. This is consistent with an earlier study which recorded neuronal ensemble activities from the STN in patients with Parkinson's disease, and which

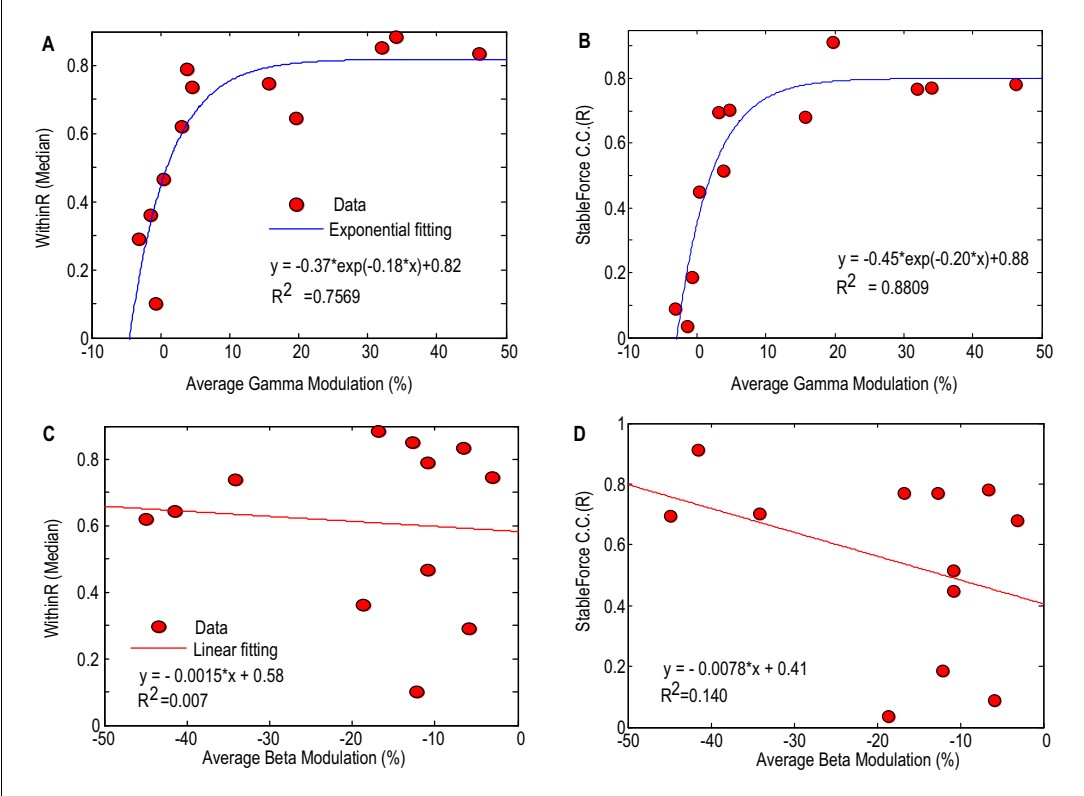

**Figure 6.** Factors affecting the gripping force prediction performance. The median values of the WithinR (**A**) and stable force correlation coefficients (**B**) increased with the average movement- related modulation in the gamma band. Each dot is the data for one STN and the blue line shows the exponential fit of the data ($y = a \cdot e^{-bx} + k$, p<0.001 for the fitting). The median values of the WithinR (**C**) and stable force correlation coefficients (**D**) show a trend of increasing with average movement related desynchronization in the beta band. The blue lines show a linear fitting, but the fits were not significant. In this Figure, we consider all the STNr with significant movement-related modulations, whether either in one or other, or both frequency bands of interest (N = 12).

showed that a large population of STN neurons were modulated by gripping force (*Patil et al., 2004*). Moreover, here we show that a simple first order linear dynamic model with the frequency-specific power change from the STN LFP as inputs is sufficient to capture the relationship between LFP signals and generated force, and can be used to decode the temporal profile of gripping force. This extends the earlier observation that frequency-specific LFP activities in the STN correlate with force-related variables in manual grips (*Anzak et al., 2012*; *Tan et al., 2013*); and that beta and gamma activities can be considered complementary non-linear correlates of force in gripping, and when combined, afford a measure that linearly correlates with force across all effort levels. Compared to other more data-driven methods based on Wiener filter for decoding force (*Flint et al., 2014*), the first order linear dynamic model proposed here simulates how the musculoskeletal plant responds to the control signal from the brain. This offers more insight in to how the basal ganglia encodes gripping force, and provides a framework to further investigate and explain the pathophysiology of motor impairment in Parkinson's disease. One further difference between the model proposed here and the Wiener filter based decoding algorithm is that the present model describes how the output from the musculoskeletal plant, i.e. the generated force, responds to the instantaneous change in the control input from the brain. This may lead to more noisy prediction in the plant output, but allows for fast behavioural reactivity in the face of movement perturbations, and therefore may represent a biologically more relevant control strategy.

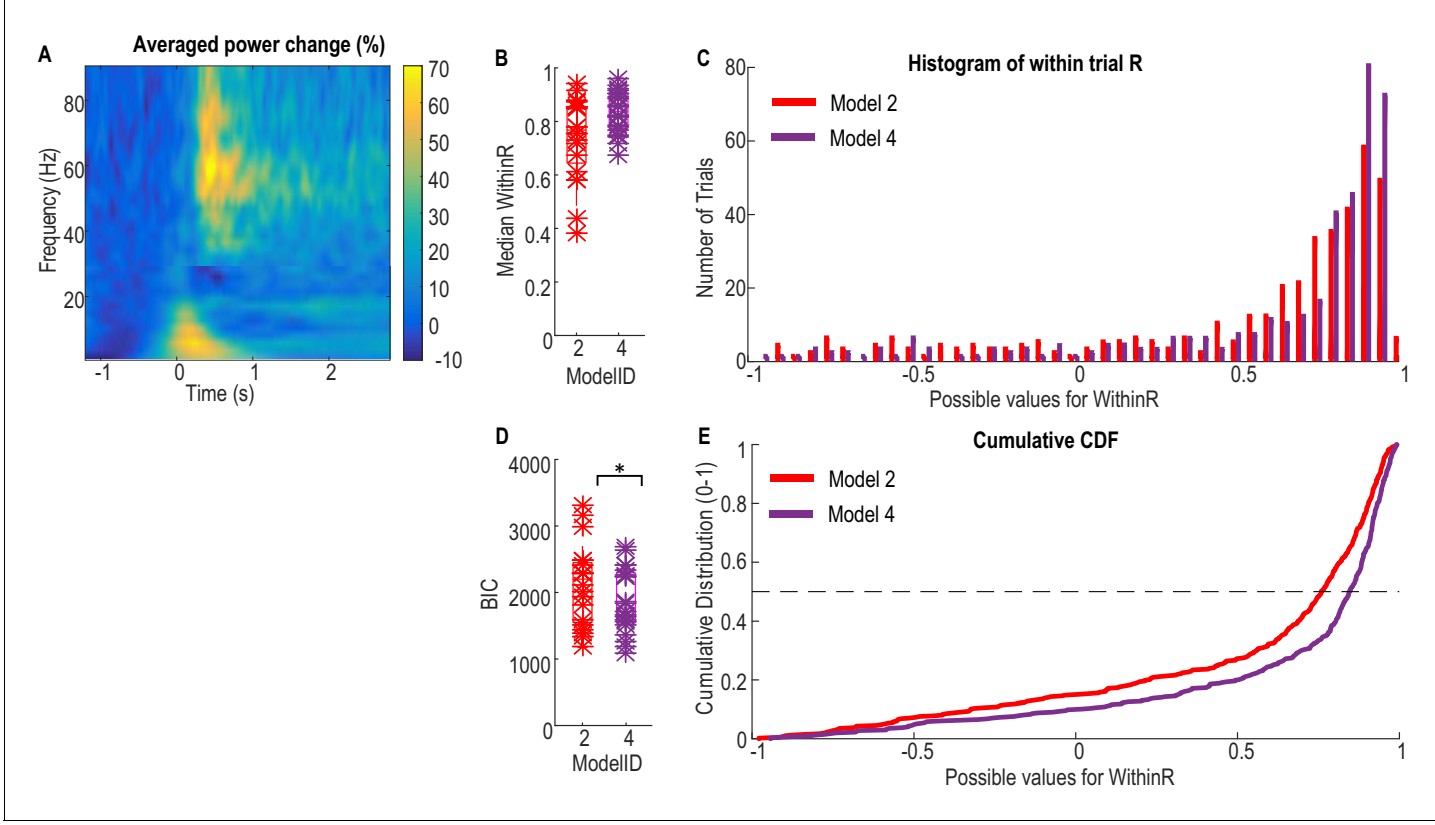

**Figure 7.** Validation of the models for force prediction on an independent patient group during maximal effort gripping. (A) Average power change in the STN LFP activity associated with the gripping movement. The power change is relative to the average over a 1 s period pre-cue. Time 0 is the timing of cue onset. (B) Median WithinR for individual STN and contralateral hands, * indicates data for each individual STN and contralateral hand. (C) Histogram and (E) cumulative distribution function (CDF) of the WithinR for all the 397 individual trials across all the 20 STNs. (D) BIC analysis showed that *Model 4* considering alpha, beta and gamma power changes significantly improved force prediction compared to *Model 2* during maximal effort gripping. * indicates p<0.05 using a paired t-test.

The following source data is available for figure 7:

**Source data 1.** The Matlab data file containing source data related to *Figure 7* (data from an independent patient group on a maximal effort gripping paradigm).

## Physiological implications: gripping force representation in STN LFPs

When gripping was performed over a range of efforts we found that most information about the gripping force profile was contained in the beta and gamma bands in the STN LFP. The rich information in the gamma band was consistent with prior studies of grasp decoding using ECoG from human motor cortex (*Flint et al., 2012*, *2014*; *Pistohl et al., 2012*). Changes in beta activity also contributed to the prediction of the force profile and inclusion of beta activity improved the encoding accuracy. This is consistent with our previous finding that beta desynchronization can encode gripping force, especially at the low effort levels (*Tan et al., 2013*, *2015*). It is unlikely that the predictive power of activities in the beta and gamma band was related to contamination by movement artefacts as we used bipolar LFPs for the decoding in which any common artefact is removed through common mode rejection. In addition, we observed increased activity in the low frequency (delta and alpha bands), but features extracted from these frequency bands deteriorated the decoding of force during our core paradigm, even though separate parameters in a first order linear model were estimated for the low frequency activities. This contrasts with previous research which showed that the local motor potential and delta activity (0–4 Hz) in the LFP from the hand area of the primary motor cortex contains information about muscle activity (*Flint et al., 2012*) and pinching force (*Flint et al., 2014*).

Our first order linear dynamic model was cross validated in an independent patient cohort in a paradigm in which patients were asked to grip as fast and hard as they could. In this case, though, the additional incorporation of theta/alpha activity further improved the accuracy and reliability of the force prediction. This may relate to the focus on maximal voluntary contractions and their execution as fast as possible, placing additional attentional demands on the participants. Oscillatory activity in the theta/alpha range may be involved in attentional mechanisms (for review see *Palva and Palva, 2007*). In particular, alpha activity (7–13 Hz) in the subthalamic nucleus of patients with Parkinson's disease is coherent with parieto-temporal cortical activity in a circuit that has been proposed to subserve attentional functions (*Hirschmann et al., 2011*; *Litvak et al., 2011*). Another consideration is the potential for the predictive power of low frequency activities to be related to contamination by movement artefacts secondary to movement overflow during maximal contractions.

Taken as a whole our results suggest that reciprocal changes in synchronised oscillatory activity in the STN can potentially provide control signals for the motor plant. In line with this is the impaired effort scaling in subjects lacking beta and gamma related motor reactivity in the STN LFP. We speculate that the motor plant may behave as a first order linear dynamical system translating a basal ganglia effort signal to force for a given effector. The present analyses allow quantitative assessment of the importance of such a simple transfer function model of basal ganglia-motor function in terms of its remarkable ability to predict both the pattern with which force is developed and the static force achieved on a single trial basis. Note that although the prediction is being used here in terms of its statistical meaning of accounting for the variance in a second signal, it also satisfies the physiological implications of this term in that changes in the STN LFP preceded changes in the measured force by about 200–300 ms. Our quantitative approach also allowed us to demonstrate that effort encoding is relatively lateralised in the basal ganglia.

## Implications for motor impairment in Parkinson's disease

Patients with Parkinson's disease have been shown to have attenuated power modulation in both the beta band (*Doyle et al., 2005*; *Devos and Defebvre, 2006*; *Androulidakis et al., 2007*; *Anzak et al., 2012*) and the gamma band (*Androulidakis et al., 2007*) during movement initialization and when a constant force is meant to be sustained (*Tan et al., 2013*) when off dopaminergic medication. Such a diminished range of power modulation in the beta and gamma bands during movement may restrict the dynamic range of the coding of force in gripping in Parkinson's disease. This may impair the force generation, and thereby cause bradykinesia, hypokinesia and weak movements when off compared to when on medication, presuming that the relationship between the STN force-encoding signal and generated force remains fixed across drug states (*Figure 8*). This is consistent with previous observations that untreated PD patients produce normal muscle activation patterns, but that muscle activity is not adequately scaled to produce the required force (*Berardelli et al., 1986*; *Turner and Desmurget, 2010*), PD patients are weaker off dopaminergic medication or DBS (*Corcos et al., 1996*; *Alberts et al., 2004*) and that PD patients show an increased probability of selecting slow movement speeds (*Mazzoni et al., 2007*). On the other hand, if the range of the maximal force to be achieved is to be kept similar to normal when off medication, the scale between the STN force-encoding signal and gripping force will be steeper when untreated (as indicated by *Figure 8B*). This will lead to abnormally high grip forces in Parkinsonian patients, as observed when they are asked to lift and hold an object (*Fellows et al., 1998*). Increased force scaling can also lead to the more coarse control of force and difficulty in finely tuning generated force. This is consistent with abnormally increased gripping force observed in precision grasping movements in Parkinson's disease (*Wenzelburger et al., 2002*).

## Subcortical LFP signals for BMI

Our findings suggest that the STN LFP could provide a high-performance control signal for BMI driven neuroprosthetic grasping in paralysed patients, leveraging advances in surgery for deep brain stimulation which has now become a relatively safe procedure (*Larson, 2014*). Surgical subcortical targets, such as the STN and globus pallidus (GPi), are involved in motor planning and execution. Activities from STN and GPi have been shown to correlate with movement parameters such as movement amplitude and speed (*Brücke et al., 2012*; *Joundi et al., 2012*), and are also modulated by movement intention (*Kühn et al., 2006*). Basal ganglia output has been theorized to regulate

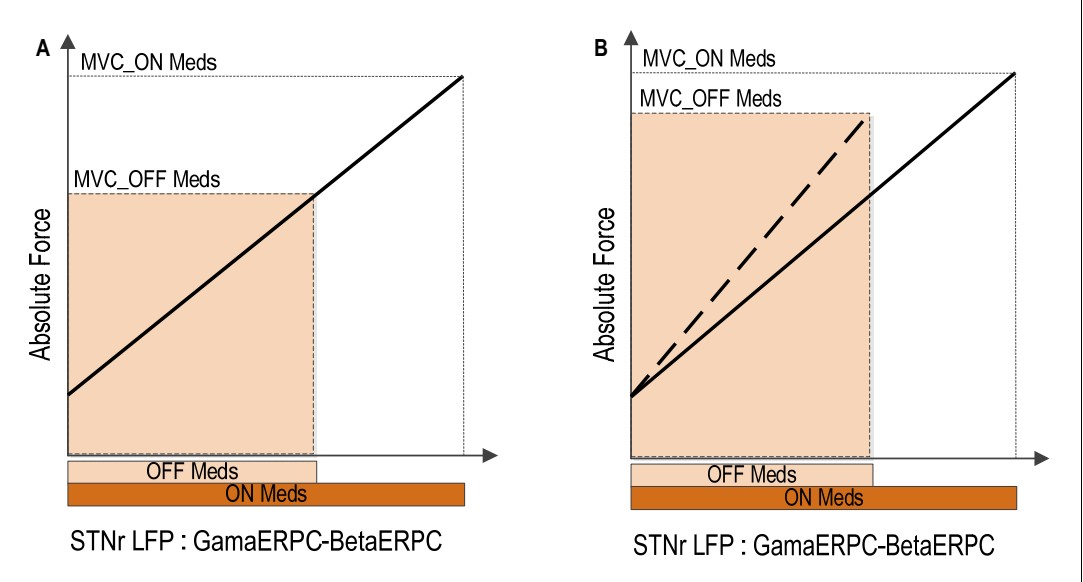

**Figure 8.** Implications of reduced movement related modulation in beta and gamma band activity in STN LFP with reduced dopaminergic input. (A) The range of forces that can be generated will be reduced if the scale between the STN encoding signal and the force is to remain the same. This will lead to unscaled, bradykinetic force generation. (B) The scale between the STN encoding signal and the force will be increased if the range of force that can be generated is to be kept similar. This will lead to abnormally high force generation and more coarse force control.

movement gain in healthy motor control, and can contain important information about the motor vigour (*Shadmehr and Krakauer, 2008*; *Turner and Desmurget, 2010*). Recordings of STN LFP signals have been shown to be stable over months using an implanted amplifier (*Quinn et al., 2015*; *Neumann et al., 2016*), and the signals are similar at re-operation several years later (*Giannicola et al., 2012*). It is also possible that the predictive potential of the STN LFP might be further improved in patients without Parkinson's disease, as dopamine increases the reactivity of beta and gamma activities around the time of movement (*Doyle et al., 2005*; *Androulidakis et al., 2007*). Moreover the reliability of predictions was dependent on the relative scale of movement-related power changes in the gamma band which could be improved by refinements in electrode design and targeting, perhaps using this reactivity to select the optimum implantation target. Therefore, LFP signals recorded from the basal ganglia have the potential to provide stable and high-performance BMI input signals for the control of neuroprosthetic devices in paralysed patients, complementary or alternative to spikes or ECoG signals recorded from the cortex. In this regard, it is important to note that some LFP reactivity is retained when movements are imagined rather than actioned (*Kühn et al., 2006*), suggesting that peripheral afferents may not be necessary for the beta suppression and gamma increases tracked here.

Nevertheless, several critical issues need to be addressed before neuroprosthetic control by STN LFPs can be considered for implementation in BMIs. Any application is predicated on the assumption that the spectral reactivity demonstrated in the STN of treated PD patients is more-or-less preserved in chronically paralysed patients. This remains to be proven. Although our results suggest that basal ganglia LFP changes might potentially be useful in force control, the selection of movements or effectors is also important in BMI control. Whether and to which extent movements involving different body parts, such the lower as opposed to the upper limb, can be decoded from the STN LFP remains unaddressed, although microelectrode recordings suggest some spatial segregation in activity related to different limbs in the human STN (*Rodriguez-Oroz et al., 2001*; *Theodosopoulos et al., 2003*). In addition, we have not demonstrated the specificity of STN LFPs for the decoding of force. Finally, plasticity of cortical control signals is likely to play a significant role in the maximisation of the performance of BMIs (*Carmena et al., 2003*; *Ganguly and Carmena, 2009*), and yet whether subcortical signals can adapt over time remains to be seen.

## Limitations and conclusion

There are some limitations in the current study which need to be acknowledged. First and foremost, we were unable to predict the force in about half of our patients. There may be many reasons for this, including disease related impairments despite dopaminergic therapy, post-operative stun effects and failure to pick up LFP activity from the 'motor' STN due to electrode targeting error. Confirmation that some electrode contacts were in or touching the STN was given by the surgical team at each centre upon review of pre- and post-operative imaging blinded to the electrophysiological results. According to this standard seven out of the nine electrodes affording force prediction were on target, with the other two electrodes in the posterolateral tail of STN. This suggests that significant movement-related spectral modulation in the beta and gamma bands, -the basis for selecting out this group in the first place, might be a good electrophysiological marker of proximity to the STN, and perhaps to the region of the STN involved in the motor representation of the hand in particular. Noteworthy in this regard, the spatial gradient of gamma reactivity was more acute than that of beta reactivity. In contrast, six out of the 12 electrodes that did not allow reasonable force prediction were identified as neither being in nor touching the STN and these subsequently revised on four sides or left inactive on two sides. A stun effect or disease related impairment might help account for the lack of joint beta and gamma reactivity in the remaining patients in this group, although two of these still had significant modulation in the beta band and one in the gamma band.

Such factors are also likely to have contributed the variation in prediction performance between subjects in whom some force prediction was possible. Amongst these, predicted force based on the STN LFP features was also noisier than the measured force. This could be caused by dynamic fluctuations and short-time-scale events in the STN LFP oscillatory activity (*Feingold et al., 2015*). It could be improved by incorporating a time history of the LFP signals which is equivalent to filtering the control input from the brain, and by incorporating filtering algorithms in the predicted force (the plant output of the model) in the time domain, such as the Kalman filter, or by convolving the output with a static nonlinearity (*Fagg et al., 2009*; *Flint et al., 2014*). Alternatively the relatively noisy predictions might in part reflect involuntary dyskinetic movement that did not impinge on grip force and yet may have been parameterised in the LFP control signal. Second, in a minority of trials grip forces were predicted based on the STN LFP but there was no measured force. This might represent a movement intention with consequent LFP change without movement execution, or voluntary movements that were not captured by the force dynamometer. Likewise these factors might have modified baseline LFPs in some trials leading to the negative force predicted in rare trials, unless predictions were clamped so as not to fall below zero. Third, the force prediction presented here focuses on the rest, force initialization, development and stable force holding phase of a grip, but does not include the force release or termination phase.

Finally, despite the observation that the first order linear model provides a good approximation of the relationship between STN LFP features and gripping force, the model may be an over simplification. Moreover, the results do not settle the discussion as to which circuits predominantly account for the selection of motor scaling, which can equally be attributed to cortical function (*DeLong and Wichmann, 2010*).

Despite these limitations, our findings do suggest that signals elaborated in and/or transmitted through the basal ganglia, and the STN in particular, carry information about motor scaling. We have also shown that features in the STN LFP combined with a simple dynamic model can be used to reliably predict the gripping force profile of the contralateral hand, even in individual grips. We propose that the LFPs from deep brain structures such as the STN could potentially provide stable and high-performance BMI input signals, complementary or alternative to neuronal spikes or ECoG signals recorded from the cortex. The recording stability and rich information content in the STN LFP about movement intentions and parameters make it an interesting signal with respect to BMI control.

## Materials and methods

The core paradigm and most subjects are the same as those in a previously published study (*Tan et al., 2013*), with two more subjects being recruited for the current study.

## Subjects

In the main paradigm, eleven patients with idiopathic Parkinson's Disease (mean disease duration 11.3 years, mean age 61.3 years, range 49–73 years; seven males) provided informed consent to take part in this study, which was approved by the local ethics committees. Patients underwent bilateral implantation of DBS electrodes into the STN, as a prelude to therapeutic high frequency stimulation for advanced idiopathic PD with motor fluctuations and/or dyskinesia. Techniques to target and implant electrodes in the STN have previously been described (*Foltynie and Hariz, 2010*). Microelectrode recordings were not made during surgery. The permanent quadripolar macroelectrode used was model 3389 (Medtronic Neurologic Division, Minneapolis, MN, USA) featuring four platinum-iridium cylindrical surfaces. Its contacts are numbered 0, 1, 2, and 3, with 0 being the most caudal and contact three being the most cranial. Localisation was supported intra-operatively by the effects of direct stimulation and by immediate post-operative stereotactic imaging. Nonetheless, in acknowledgement of the fact that not all electrode contacts could be expected to lie in the STN per se, we term the area sampled by the electrode contact the STN region (STNr). DBS electrode extension cables were externalized through the scalp to enable recordings prior to connection to a subcutaneous DBS pacemaker, implanted in a second operative procedure up to seven days later. One out of the eleven patients (case 2) had only one electrode externalised for testing, thus we could record from 21 STN regions (STNr). Clinical details of the patients are given in *Table 1*. The patients showed 53.4 ± 6.0% (p<0.001) improvement in the motor section of the Unified Parkinson's Disease Rating Scale (UPDRS) on treatment with levodopa, indicating good responsiveness to this drug.

## Experimental paradigm

Subjects were seated in a comfortable chair with their shoulders adducted and their elbows flexed at about 90°. Subjects were first asked to grip the dynamometer with maximal effort three times, with each trial lasting for 3 s. Then they were presented with a series of imperative visual cues (red light-emitting diode illuminated for 3 s), separated by 11–13 s, and instructed to 'choose an effort level from the scale provided and then to squeeze the force dynamometer at this chosen effort level when the light comes on and maintain this squeeze for the duration of the light'. Subjects were provided with the Rated Perceived Exertion Scale with 11 levels ranging from zero to 10 (*Borg, 1998*) printed on a piece of A4 paper. They were asked to try and randomise their selection of effort levels, so that the levels were varied from trial to trial, and all levels were represented. The subjects reported the effort level verbally after each grip. The mean number (± SEM) of trials per hand per subject was 31 ± 2 grips, with a mean number of trials per level per subject of 3 (±1). In particular, 2–3 trials were self-rated as maximal effort and 2–3 trials self-rated as minimal effort in each subject. Patients were asked to grip following illumination of the LED, but were not requested to respond as quickly as possible. There was no other feedback provided to the patients related to the generated force.

## Recordings

Recordings were made when the patients were ON their usual dopaminergic medication, 3–6 days postoperatively, while electrodes were externalized and before implantation of the pulse generator. Grip force was measured one hand at a time using an isometric dynamometer with standard Jamar design, and its handle set in the second of the five discrete grip diameter adjustments possible (G200; Biometrics Ltd, Gwent, UK). The order in which left and right hands were tested was counterbalanced across subjects. Monopolar LFPs recorded with a TMSi porti (TMS international, Netherlands) and its respective software. A common average reference was used for the monopolar recordings and these were low and high pass filtered at 0.5 and 500 Hz, respectively. Bipolar signals were derived offline by subtracting the monopolar recordings between neighbouring contacts on each electrode. The force was only a low pass filtered at 200 Hz. EMG signals from the first dorsal interosseous (FDI) of the activated hand were recorded in all patients, and the EMG signals from the extensor digitorum communis or extensor carpi ulnaris of both lower arms were recorded in six out of the 11 patients. LFP, EMG and force measurements were initially sampled at 2048 Hz. The effort level the subject reported verbally after each grip was logged manually and then used to label each individual trial.

## LFP analysis

Each electrode has four contact points, and the LFP data were converted off-line to give three bipolar contact pairs (01, 12 and 23) per electrode. Nonetheless, in acknowledgement of the fact that not all electrode contacts could be expected to lie in the STN per se, we term the area sampled by the electrode contact the STN region (STNr). Continuous wavelet transform, with Morlet wavelet and cycle number of 7, was then applied to LFP recordings from each bipolar contact pairs for time-frequency decomposition. The average power changes relative to the pre-movement baseline over the three trials of maximal effort gripping was calculated for each bipolar contact in the contralateral STNr. Three features from each bipolar LFP signal were extracted: the power change in the theta/alpha (4–12 Hz) band, beta (13–30 Hz) and gamma (55–90 Hz) frequency bands. For each electrode, the bipolar signal with the largest movement related reduction in the beta power was selected for analysis. Further, the average power in each frequency band and at each time point was compared against the distribution of the average power for that frequency over a one second period of time before the cue. Significant movement-related modulations were defined as those trials in which there were at least 50% of time points during the second after movement onset with power smaller than the 5% boundary (to capture event related desynchronization) or larger than the 95% boundary (to capture event related synchronisation) of the power distribution before movement for that frequency band. This procedure identified 9 STNs (from six different patients) with significant movement-related modulations in both the beta band and the gamma band; three more STNs showed significant movement-related modulations in either the beta or gamma band. However, there was no significant movement-related modulation in either beta or gamma band in the remaining 9 STNs. Details of the patient and the average power changes during the second after cue onset for maximal effort gripping in the beta and gamma bands for each STN are presented in *Table 1*.

For force decoding, time-frequency decomposition using continuous wavelet transform was applied to the STN LFPs from the bipolar contact previously selected. For each individual trial of gripping, the power change for each frequency at each time point was calculated by normalizing the power at that time point against the average power during the 1 s before cue presentation, so 0 indicates the power being the same as the baseline activity before cue, positive values indicate power increase (referred to as ERS) and negative value indicate power decrease (referred to as ERD). Then average power changes in theta/alpha ($\alpha$: 4–12 Hz) band, the beta ($\beta$: 13–30 Hz) and gamma ($\gamma$: 55–90 Hz) frequency bands at each time point were calculated. The latter frequency range of 55–90 Hz was selected on the basis of our previous study showing that STN LFP activities within this range increase during the onset of a grip (*Anzak et al., 2012*) and correlate with the stable force achieved during a grip (*Tan et al., 2013*).

## Force decoding procedure

Our previous study (*Tan et al., 2013*) showed that activity changes in the beta and gamma bands make major contributions to the encoding of efforts in gripping. In the present study, we tested different hypotheses about the dynamic relationship between LFP features and force. Based on the results from the previous study showing that the difference between gamma ($\gamma$) and beta ($\beta$) modulations correlates with effort at the force holding phase, the first model to be tested used the signal of $\gamma - \beta$ as the as a control input:

$$Force = (\gamma - \beta) * \frac{K_p}{T_p \cdot s + 1} e^{-T_d \cdot s} \tag{Model 1}$$

Where $\beta$ and $\gamma$ are beta and gamma band activity change in the STNr LFP, respectively. $\frac{K_p}{T_p \cdot s + 1} e^{-T_d \cdot s}$ is a standard representation of a first order linear dynamic system with a time delay in the Laplace domain, where $K_p$ is the steady state proportional gain, $T_p$ is the time constant which is a measure of how fast the force output responds to brain signal changes, and $T_d$ is the time delay between the brain control signal (LFP) and the measured force which describes the latency between the timing of brain signal changes and force output onset, $s = j \cdot \omega$ which is a complex variable. The equivalent of *Model 1* in the time domain is: $Force(t) + T_p \cdot \frac{\partial Force(t)}{\partial t} = K_p \cdot (\gamma(t - T_d) - \beta(t - T_d))$, which suggests that beta and gamma modulations in STN LFP encode the instantaneous amplitude and the differentiation of force over time.

**Table 1.** Patient details and movement-related modulated in beta and gamma bands.

| Patient ID | Age (yrs) | Gender | PD duration (yrs) | Main symptoms | Daily dose (mg) | UPDRS part III (Pre-op) OFF | ON | Left STN Bipolar channel | Left Beta ERD | Left Gamma ERS | Left Localisation | Left Most % beta | Left Stim setting | Right STN Bipolar channel | Right Beta ERD | Right Gamma ERS | Right Localisation | Right Most % beta | Right Stim setting | Stimulation effect |
|---|---|---|---|---|---|---|---|---|---|---|---|---|---|---|---|---|---|---|---|---|
| 1 QS | 49 | M | 13 | Stiffness, bradykinesia, bilateral tremor, freezing | Levodopa 800 Apomorphine (6.5 mg/hour) Rotigotine 8 | 38 | 13 | L1L2 | −4.52 (*) | 17.99 (*) | L1,L2 border MED | L1 | Case: + L1: - | R1R2 | 1.15 | 33.2 (*) | R0:inside/border/MED; R1: border/MED | R0 | Case: + R1: - | UPDRS OFF Med, Stim ON/OFF: 13/38 |
| 2 Ox | 69 | M | 11 | Rigidity, bradykinesia, freezing | Ropinirole 8 Pramipexole 0.75 Levodopa 900 | 38 | 18 | | – | – | Electrode was not in target and therefore not recorded | | | R0R1 | −4.79 | 0.92 | R0,R1 inside only | R1 | None | Stimulation was discontinued shortly after surgery due to unsatisfactory clinical effect |
| 3 King | 65 | F | 17 | Rigidity, tremor | Amantadine 400 Levodopa 600 | 55 | 49 | L0L1 | −5.12 (*) | 7.44 (*) | All inside | L1 | Case: + L0: - | R0R1 | −29.03 (*) | 3.29 (*) | All inside | R1 | Case: + R1: - | Not evaluated |
| 4 QS | 56 | M | 10 | Bradykinesia, rigidity, tremor limping gait | Levodopa 1000 Rasagiline 1 Citalopram 20 | 40 | 12 | L1L2 | −2.48 (*) | 37.73 (*) | L1,L2,L3 inside; L2 dorsolat | L2 | Case: + L1: - | R0R1 | −18.92 (*) | 2.98 (*) | R0 inside; R1 border/ dorsolat | R1 | Case: + R0, R1 (alternating): - | UPDRS OFF Med, Stim ON/OFF: 29/40 |
| 5 QS | 60 | F | 11 | Tremor@Left; poor coordination, bended gait | Levodopa 600 Pramipexole 0.75 | 53 | 16 | L1L2 | −4.94 (*) | 6.68 (*) | All inside; L1 dorsolat | L1 | Case: + L1: - | R2R3 | −0.049 (*) | 2.14 | R1 inside; R2 border | R2 | Case: + R1: - | Not evaluated |
| 6 Kings | 65 | M | 5 | Rigidity, bradykinesia, motor fluctuation, tremor | Levodopa 400 Entacapone 800 Rotigotine 8 | 41 | 29 | L1L2 | 3.33 | 1.81 | All inside | None | Case + L2: - | R1R2 | −7.37 (*) | 0.66 | All inside | R2 | Case + R1: - | Not evaluated |
| 7 QS | 56 | M | 10 | tremor@all four limbs | Levodopa 600 Rotigotine 8 Seleginine 10 | 52 | 19 | L0L1 | −10.68 (*) | 8.42 (*) | L2, L3 in superior STN | L0 | Case: + L1: - | R0R1 | −22.76 (*) | 14.02 (*) | R0, R1 in STN, R2 lateral border of superior STN | R1 | Case: + R1: - | Relocation after recording due to side effects on speech |
| 8 Kings | 73 | M | 14 | Bradykinesia, tremor | Rotigatine 16 Selegeline 10 Levodopa 700 | 35 | 15 | L0L1 | 0.157 | −0.186 | All inside | None | Case: + L1: - | R1R2 | −4.93 (*) | 5.57 (*) | All inside | R1 | Case: + R1: - | Not evaluated |
| 9 Ox | 63 | F | 14 | Rigidity, bradykinesia | Ropinirole 23 Levodopa 150 | 35 | 24 | | 3.197 | −1.14 | None inside | None | None | | −2.59 | 7.20 | None inside | None | None | Post-op imaging show mis-location, and electrodes were relocated to GPi |
| 10 QS | 66 | F | 16 | Shuffle, poor balance, NO tremor | Levodopa 600 Amantadine 200 Ropinirole 24 Rasagiline 1 | 32 | 13 | L0L2(L1 no signal) | 4.33 | 2.95 | L0,L1 inside | L0 | Case: + L1: - | R0R1 (bipolar reduced modulation) | −1.37 | 7.41 (*) | R1,R2 inside | R1 | Case: + R1: - | UPDRS OFF Med, Stim ON/OFF: 26/32 |
| 11 QS | 52 | M | 7 | Freezing, falls, postural instability, tremor@right side | Levodopa 1300 Citalopram 20 Trihexyphenidyl 6 | 58 | 13 | L1L2 (bipolar reduced modulation) | 38.77 | 13.22 | | L2 | Case: + L1: - | R1R2 (bipolar reduced modulation) | −1.05 | 1.11 | | R1 | Case: + R1: - | Relocation after recording |
| Mean | 61.3 | | 11.3 | | | 43.4 | 20.1 | | | | | | | | | | | | | |

(*) Indicate significant movement-related modulation in the power of the activity of the specific frequency band; Ox, Kings, QS indicate the three neurosurgical centres where the data were recorded: Ox = John Radcliffe Hospital, University of Oxford; Kings = Department of Neurosurgery, Kings College Hospital, Kings College London; QS = Sobell Department of Motor Neuroscience and Movement Disorders, UCL Institute of Neurology.

**Table 2.** Model details.

*Models 1–3* use activity change in the beta band ($\beta$) and gamma bands ($\gamma$) from the STN LFP as model inputs. *Models 4 and 5* take into account extra information about the low frequency activity change ($\alpha$). *Models 6–8* only use the activity change from a single frequency band ($\alpha$, $\beta$ and $\gamma$, respectively) as model input. $T_p$ and $T_d$ are the time constant and time delay of the first order linear dynamic model, respectively.

| Model ID | Model equation | No. of free parameters |
|---|---|---|
| 1 | $Force = (\gamma - \beta) * \frac{K_p}{T_p \cdot s + 1} e^{-T_d \cdot s}$ | 3 |
| 2 | $Force = \left(K_{p1} \cdot \gamma + K_{p2} \cdot \beta\right) * \frac{1}{T_p \cdot s + 1} e^{-T_d \cdot s}$ | 4 |
| 3 | $Force = \gamma * \frac{K_{p1}}{T_{p1} \cdot s + 1} e^{-T_{d1} \cdot s} + \beta * \frac{K_{p2}}{T_{p2} \cdot s + 1} e^{-T_{d2} \cdot s}$ | 6 |
| 4 | $Force = \left(K_{p1} \cdot \gamma + K_{p2} \cdot \beta + K_{p3} \cdot \alpha\right) * \frac{1}{T_p \cdot s + 1} e^{-T_d \cdot s}$ | 5 |
| 5 | $Force = \gamma * \frac{K_{p1}}{T_{p1} \cdot s + 1} e^{-T_{d1} \cdot s} + \beta * \frac{K_{p2}}{T_{p2} \cdot s + 1} e^{-T_{d2} \cdot s} + \alpha * \frac{K_{p3}}{T_{p3} \cdot s + 1} e^{-T_{d3} \cdot s}$ | 9 |
| 6 | $Force = \alpha * \frac{K_p}{T_p \cdot s + 1} e^{-T_d \cdot s}$ | 3 |
| 7 | $Force = \beta * \frac{K_p}{T_p \cdot s + 1} e^{-T_d \cdot s}$ | 3 |
| 8 | $Force = \gamma * \frac{K_p}{T_p \cdot s + 1} e^{-T_d \cdot s}$ | 3 |

The second model to be tested assumed that the activities from beta and gamma bands have a different proportional gain, but have the same dynamic relationship in terms of time constant ($T_p$) and time delay ($T_d$) in encoding force:

$$Force = \left(K_{p1} \cdot \gamma + K_{p2} \cdot \beta\right) * \frac{1}{T_p \cdot s + 1} e^{-T_d \cdot s} \qquad \text{(Model 2)}$$

The third model tested assumed that the dynamic relationship between force and activities at different frequency bands are different, with different values for the proportional gain ($K_p$), time constant ($T_p$) and time delay ($T_d$). Thus the generated force is the sum of the distinct processes with different inputs and different transfer function parameters (*Model 3*):

$$Force = \gamma * \frac{K_{p1}}{T_{p1} \cdot s + 1} e^{-T_{d1} \cdot s} + \beta * \frac{K_{p2}}{T_{p2} \cdot s + 1} e^{-T_{d2} \cdot s} \qquad \text{(Model 3)}$$

These models were compared against models which include extra information from the STNr LFP in the form of the relative power change in the theta/alpha frequency band ($\alpha$) (*Model 4* and *Model 5*):

$$Force = \left(K_{p1} \cdot \gamma + K_{p2} \cdot \beta + K_{p3} \cdot \alpha\right) * \frac{1}{T_p \cdot s + 1} e^{-T_d \cdot s} \qquad \text{(Model 4)}$$

$$Force = \gamma * \frac{K_{p1}}{T_{p1} \cdot s + 1} e^{-T_{d1} \cdot s} + \beta * \frac{K_{p2}}{T_{p2} \cdot s + 1} e^{-T_{d2} \cdot s} + \alpha * \frac{K_{p3}}{T_{p3} \cdot s + 1} e^{-T_{d3} \cdot s} \qquad \text{(Model 5)}$$

Models with only activities from single frequency bands were also evaluated to see if the combination of activities from different frequency bands was necessary for the decoding for force:

$$Force = \alpha * \frac{K_p}{T_p \cdot s + 1} e^{-T_d \cdot s} \qquad \text{(Model 6)}$$

$$Force = \beta * \frac{K_p}{T_p \cdot s + 1} e^{-T_d \cdot s} \qquad \text{(Model 7)}$$

$$Force = \gamma * \frac{K_p}{T_p \cdot s + 1} e^{-T_d \cdot s} \qquad \text{(Model 8)}$$

The predicted force was clamped so it did not fall below zero. The parameters in different models ($K_p$, $T_p$, $T_d$) were estimated for each STN separately.

### Cross validation and model assessment

Cross-validation was used to evaluate the generalisability and decoding performance of the proposed models with the STNr LFP features as inputs for predicting gripping force. For each STN/contralateral hand, the parameters of different models were first identified using least-squares optimisation applied on a set of training data to fit the corresponding STNr LFP features and measured force of the training data. The decoding accuracy of the models was evaluated by applying the model with identified parameters on another set of testing data. When evaluating the performance of the model applied to across-trial averages, the average of STNr LFP features and forces over multiple trials of low effort levels was first used as the training data to identify the model parameters, and the model was then tested on average data from high effort trials; or vice versa, i.e. using the across-trial average of high effort trials as training data and the averages from low effort trials as testing data. Bayesian information criterion (BIC) was used for model selection. The BIC value for each model is calculated as: $BIC = n \cdot ln(\sigma_e^2) + k \cdot \ln(n)$; where $\sigma_e^2$ is the error variance: $\sigma_e^2 = \frac{1}{n} \cdot \sum_{t=1}^{n} \left( F(t) - \hat{F}(t) \right)^2$ with $F(t)$ and $\hat{F}(t)$ the actually measured and predicted force of each time point respectively; n is the total number of time points; k is the number of free parameters in the model.

When evaluating the performance of the model on individual trials, a five-fold cross validation was used. All data from each recording session were partitioned into five equal folds. During each iteration, one fold was retained for testing, and the other four folds were used as training data to identify model parameters. Five iterations would allow for each observation being used for validation exactly once. The five results were then combined to produce a single complete estimation for one session. Different evaluation parameters including within trial correlation (WithinTrialR), correlation between the measured and predicted stable force were quantified based on the testing data.

In addition, for cross-subject validation, the 1st order dynamic model with STN LFP features as inputs was used to predict the gripping force of individual trials in another independent patient group. Ten patients were recorded in the study when the patients were asked to grip as fast and as hard as they can in each trial in response to an external cue. The details of patient information and experimental paradigms were reported in a previous publication (*Anzak et al., 2012*). Twenty trials were collected for each hand when the patients were on their normal dopaminergic medication. Within each STN in this independent patient group, a 4-fold cross validation was used. For each iteration, 15 trials were used to fit the model to estimate model parameters and the model was used to predict force on the remaining five trials. The same procedure was repeated four times so each trial was used for prediction exactly once. The correlation coefficients between the predicted force and measured force for each individual trial were quantified.

All analyses were performed in Matlab (version 2012b). Median and the range of values are reported if the sample number is smaller than 10, or the distribution is not normal. Otherwise, means ± standard error of means (SEM) are presented throughout the text. Correlation coefficients were Fisher-z transformed before any statistical test, but the raw values were presented in the text and the figures.

## Acknowledgements

This work was funded by the EU grant FP7-ICT-610391, Medical Research Council, Rosetrees Trust, and the National Institute of Health Research, Oxford Biomedical Research Centre.

# Additional information

## Funding

| Funder | Grant reference number | Author |
|---|---|---|
| European Commission | FP7-ICT-610391 | Huiling Tan<br>Alek Pogosyan<br>Peter Brown |
| Medical Research Council | Unit Grant | Huiling Tan<br>Alek Pogosyan<br>Peter Brown |
| National Institute for Health Research | Oxford Biomedical Research Centre | Peter Brown |
| Rosetrees Trust | | Peter Brown |

The funders had no role in study design, data collection and interpretation, or the decision to submit the work for publication.

## Author contributions

HT, PB, Conception and design, Acquisition of data, Analysis and interpretation of data, Drafting or revising the article; AP, Acquisition of data, Analysis and interpretation of data, Drafting or revising the article; KA, ALG, TA, LZ, Acquisition of data, Drafting or revising the article; TF, PL, MH, Acquisition of data

## Ethics

Human subjects: Informed consent and consent to publish was obtained from patients before they took part in the study, which was approved by Oxfordshire Research Ethics Committee.

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
