## [Decision Letter]

Thank you for submitting your article "Decoding gripping force based on local field potentials recorded from subthalamic nucleus in humans" for consideration by *eLife*. Your article has been reviewed by three peer reviewers, and the evaluation has been overseen by a Reviewing Editor and Sabine Kastner as the Senior Editor. The reviewers have opted to remain anonymous.

The reviewers have discussed the reviews with one another and the Reviewing Editor has drafted this decision to help you prepare a revised submission.

Summary:

In this manuscript Tan et-al investigate the nature of the basal ganglia control signal for force. To do so they dynamically decoded gripping force based on LFP recorded from the STN in Parkinson's disease patients with implanted DBS electrodes. In nine STN (out of 21 tested) of 5 patients (out of 11 tested) they found significant movement related modulations in the beta and the gamma frequency domain. Further analysis of the activity of these 9 STN revealed that gamma (55-95 Hz) and beta (13-30 Hz) bands of the STN LFP were most informative about gripping force, and that a first order dynamic linear model can be used to decode the temporal profile of gripping force.

Essential revisions:

There was considerable agreement between reviewers on the topics that require added attention.

1) Please provide information and discussion regarding the lack of significant task-related changes in beta/gamma activity in approximately 50% of cases.

2) Provide additional information regarding the locations of electrode contacts relative to STN boundaries, the signals present on different contacts and how those signals relate to clinical efficacy of DBS through those contacts.

3) Compare the different models using AIC or BIC metrics and a more appropriate cross-validation approach.

4) Acknowledge limitations and moderate the more speculative claims regarding the utility of this signal for BMI. For example, the generality (across tasks and limbs) and specificity of the signal (to force) has not been tested.

---

## [Author Response]

*Essential revisions:*

*There was considerable agreement between reviewers on the topics that require added attention.*

*1) Please provide information and discussion regarding the lack of significant task-related changes in beta/gamma activity in approximately 50% of cases.*

We thank the reviewers for their comments. With extra analysis, we have now identified two correlates which can contribute to the lack of significant task-related changes in approximated 50% of cases.

First of all, further analyses on the measured force have shown significant differences in how force scales with effort between a first group with movement-related beta and gamma band reactivity, and a second group without such reactivity. We have now included the average force trajectories of different effort levels for the two patient groups in Figure 1 (Figure 1). We have also calculated the Spearman correlation coefficient between the average force during the holding phase (1.0-2.0s after cue) and the self-rated effort (SRE) for each hand in both groups. This has identified significantly lower correlation coefficients between stable force and self-rated effort in the 2nd group (also shown in Figure 1), indicating impairment in the scaling of force with effort in the 2nd group. The following has been included in the Results: “Figure 1 show the normalized force trajectories with different self-rated effort (SRE) levels averaged across hands in the two groups. Figure 1 show the trajectories of force yank (the differentiation of force against time) with different SRE for the two groups. […] The first group of 9 STN sides showing significant movement-related modulations in both bands were selected for further analysis as (1) this group showed normal scaling of force and force yank with effort; (2) movement-related modulations were presumed to indicate recordings made in or near the dorsal STN; (3) significant movement-related modulations were taken as evidence of relatively physiological functioning given that the motor reactivity of the two rhythms is increased by dopaminergic medication (Anzak et al., 2012), and (4) lack of movement related modulations might indicate a temporary surgical stun effect and temporary damage to the STN (Chen et al., 2006) or targeting variance.”

The above new findings are now also mentioned in the Discussion: “Taken as a whole our results suggest that STN LFP features, in particular the reciprocal changes in synchronised population activity in the beta and gamma frequency ranges in the STN can potentially provide control signals for the motor plant. In line with this is the impaired effort scaling in subjects lacking beta and gamma related motor reactivity in the STN LFP.”

In addition, the variation in the target location of the electrodes can also contribute to the variation in the oscillatory responses measured from the DBS electrodes. We have now added the following paragraph to the section in the Discussion entitled “Limitations and conclusion”: “There are some limitations in the current study which need to be acknowledged. […] Such factors are also likely to have contributed to the variation in prediction performance between subjects in whom some force prediction was possible. Amongst these, predicted force based on STN LFP features was also noisier than the measured force…”.

*2) Provide additional information regarding the locations of electrode contacts relative to STN boundaries, the signals present on different contacts and how those signals relate to clinical efficacy of DBS through those contacts.*

We have now included whether electrode contacts were in or touching the STN as classified by the surgical team at each centre upon review of pre- and post-operative imaging blinded to the electrophysiological results (Table 1). We also now include in the latter the bipolar contact pairs which revealed the reported physiological pattern in each STN, as well as the contacts that were utilised for clinical stimulation.

Unfortunately, we only have UPDRS scores ON and OFF stimulation while OFF medication in three patients, which are now included in Table 1. Three patients had electrodes re-implanted after recordings and in one more patient DBS was not used for chronic treatment (detailed in Table 1). Accordingly no relevant UPDRS scores are available in these subjects while ON and OFF stimulation. Finally, the relevant post-op assessments were not performed in four patients.

*3) Compare the different models using AIC or BIC metrics and a more appropriate cross-validation approach.*

Bayesian information criterion has now been used for model selection. The following has been included in the Methods: Bayesian information criterion (BIC) was used for model selection. The BIC value for each model is calculated as: BIC=n∙ln(σe2)+k∙ln(n); where σe2 is the error variance: σe2=1n∙∑t=1n(F(t)−F^(t))2 with F(t) and F^(t) the actually measured and predicted force of each time point respectively; n is the total number of time points; k is the number of free parameters in the model.

The BIC values for each model are now shown in Figure 2. The statistical results of the BIC analysis have been included in the Results: ‘Figure 2 shows the BIC values combining the force prediction for low effort and high effort for all models. One-way repeated ANOVA identified a significant effect of the models on the BIC (F(7,56) = 11.777, p< 0.001). Paired t-tests showed that Model 2 was significantly better, in terms of BIC, than Models 4-8 after FDR multiple comparison correction: (t(8) = -3.63, p = 0.007 compared with Model 4; t(8) = -4.950, p = 0.001 compared with Model 5; t(8) = -9.347, p < 0.001 compared with Model 6; t(8) = -5.757, p < 0.001 compared with Model 7; t(8) = -3.675, p = 0.007 compared with Model 8).’

We have now also cross-validated the models on an independent, previously published, patient group, who were asked to grip with maximal effort in each individual trial in the paradigm. The following has been included in the manuscript: “Finally, the first order dynamic model with beta and gamma power changes as inputs (Model 2) was used to predict gripping force from individual trials in an independent patient group that performed a different, but related, paradigm. Ten patients were recorded in this study and were asked to grip as fast and as strongly as they could in each trial in response to an external cue. Twenty trials were collected for each hand when the patients were on their normal dopaminergic medication. Details of patient information and experimental paradigm were previously reported (Anzak et al., 2012). The average movement related power change in the STN LFP activities and results of force prediction in this patient group are shown in Figure 7. The median correlation coefficient between predicted force and measured force for individual trials (median WithinR) ranged between 0.3848 and 0.9421 for the 20 STNs and contralateral hand (Figure 7). Considering all the 397 individual trials across all STNs, more than 50% of the trials had WithinR more than 0.7859 (Figure 7). We also found that incorporating α activity (Model 4) improved the force prediction accuracy for maximal effort gripping in this patient group. When Model 4 was used, the median WithinR ranged between 0.6731 and 0.9625 across all STNs, and more than 50% of all individual trials had WithinR more than 0.8699. BIC analysis showed that Model 4 (BIC = 2116.2 ± 135.7) was significantly better than Model 2 (BIC = 1873.0 ± 110.3) for this patient group performing maximal effort gripping (∆BIC = 243.3 ± 95.9, t(19) = 2.5369, p = 0.020 comparing the BIC values, Figure 7)”.

The following revisions have also been made in the Discussion: “When gripping was performed over a range of efforts we found that most information about the gripping force profile was contained in the beta and gamma bands in the STN LFP. […] Another consideration is the potential for the predictive power of low frequency activities to be related to contamination by movement artefacts secondary to movement overflow during maximal contractions.”

*4) Acknowledge limitations and moderate the more speculative claims regarding the utility of this signal for BMI. For example, the generality (across tasks and limbs) and specificity of the signal (to force) has not been tested.*

Potential limitations of STN LFP signals for neuroprosthetic control in BMIs have now been discussed in a new paragraph added to the section of the Discussion titled Subcortical LFP signals for BMI: “Nevertheless, several critical issues need to be addressed before neuroprosthetic control by STN LFPs can be considered for implementation in BMIs. Any application is predicated on the assumption that the spectral reactivity demonstrated in the STN of treated PD patients is more-or-less preserved in chronically paralysed patients. […] Finally, plasticity of cortical control signals is likely to play a significant role in the maximisation of the performance of BMIs (Carmena 2003; Ganguly and Carmena 2009), and yet whether subcortical signals can adapt over time remains to be seen.”

Elsewhere in the Abstract, Introduction and Discussion we have now further moderated our references to BMI interface control (e.g. “may potentially be useful” etc.).